# Preventing childhood obesity, phase II feasibility study focusing on South Asians: BEACHeS

Peymané Adab,[1] Miranda J Pallan,[1] Janet Cade,[2] Ulf Ekelund,[3,4] Timothy Barrett,[5] Amanda Daley,[1] Jonathan Deeks,[1] Joan Duda,[6] Paramjit Gill,[1] Jayne Parry,[1] Raj Bhopal,[7] K K Cheng[1]

▶ Prepublication history and additional material for this paper is available. To view please visit the journal (http://dx.doi.org/10.1136/bmjopen-2013-004579).

For numbered affiliations see end of article.

Correspondence to
Dr Miranda J Pallan;
m.j.pallan@bham.ac.uk

## ABSTRACT

**Objective:** To assess feasibility and acceptability of a multifaceted, culturally appropriate intervention for preventing obesity in South Asian children, and to obtain data to inform sample size for a definitive trial.

**Design:** Phase II feasibility study of a complex intervention.

**Setting:** 8 primary schools in inner city Birmingham, UK, within populations that are predominantly South Asian.

**Participants:** 1090 children aged 6–8 years took part in the intervention. 571 (85.9% from South Asian background) underwent baseline measures. 85.5% (n=488) were followed up 2 years later.

**Interventions:** The 1-year intervention consisted of school-based and family-based activities, targeting dietary and physical activity behaviours. The intervention was modified and refined throughout the period of delivery.

**Main outcome measures:** Acceptability and feasibility of the intervention and of measurements required to assess outcomes in a definitive trial. The difference in body mass index (BMI) z-score between arms was used to inform sample size calculations for a definitive trial.

**Results:** Some intervention components (increasing school physical activity opportunities, family cooking skills workshops, signposting of local leisure facilities and attending day event at a football club) were feasible and acceptable. Other components were acceptable, but not feasible. Promoting walking groups was neither acceptable nor feasible. At follow-up, children in the intervention compared with the control group were less likely to be obese (OR 0.41; 0.19 to 0.89), and had lower adjusted BMI z-score (−0.15 kg/m²; 95% CI −0.27 to −0.03).

**Conclusions:** The feasibility study informed components for an intervention programme. The favourable direction of outcome for weight status in the intervention group supports the need for a definitive trial. A cluster randomised controlled trial is now underway to assess the clinical and cost-effectiveness of the intervention.

**Trial registration number:** ISRCTN51016370.

### Strengths and limitations of this study

- We report the findings of a feasibility study of a childhood obesity prevention intervention that focuses on primary school-aged children from South Asian communities in the UK. Despite their susceptibility to the cardiometabolic consequences of obesity, little obesity prevention research has been undertaken in these communities previously.
- The early phases of the UK Medical Research Framework for complex health interventions have guided the intervention development and evaluation process undertaken in this feasibility study.
- The feasibility and acceptability of the childhood obesity prevention intervention components was variable and context dependent; however, the exploratory nature of the study enabled us to modify and refine delivery of the intervention throughout.
- Development and evaluation of the feasibility and acceptability of the intervention was undertaken in materially disadvantaged, predominantly South Asian communities, thus its transferability would be dependent on tailoring to the specific local context.
- The final intervention programme, following modification and refinement in this feasibility study, is being definitively evaluated in an ongoing cluster-randomised controlled trial.

## INTRODUCTION

Childhood obesity is a growing problem worldwide.[1] Apart from psychological and social problems, longitudinal studies show adverse future health consequences in children as young as 7 years old who are obese.[2] In the UK, although childhood overweight prevalence has stabilised, socioeconomic disparities have widened, with increasing trend in more deprived subpopulations.[3] Data from the national childhood surveillance programmes in England show that at school entry (age 4–5 years), 9.5% of children are

obese (ie, above 95th centile for national reference standards), but this prevalence doubles (19.2%) by the end of primary school (age 11).[4] The rate of increase among children from South Asian (SA) ethnic groups, especially girls, is greater than that for the population as a whole (increasing trend of 1.13% and 0.66% per year for Bangladeshi and Pakistani girls, respectively, compared with 0.35% yearly increase in White British).[5] Thus the primary school period presents a key phase for prevention, and SA are an important target group.

However, despite numerous systematic reviews,[6 7] reports[8 9] and guidelines,[10] evidence for effective approaches to prevention is limited, particularly among minority ethnic groups. Relevant trials suggest that multifaceted school-based interventions have potential, particularly those that also include a home or community element, but the most effective combination of components is not clear.[7 9] The need for involving stakeholders, such as families, schools and local communities, in the decision-making regarding potential intervention strategies has been highlighted.[6] Furthermore, for a complex intervention such as obesity prevention, which has several interconnecting components, a rigorous and iterative-phased approach is required to improve study design, execution and applicability of results. The UK Medical Research Council (MRC) proposed a framework for such interventions.[11] Given the growing problem of obesity and lack of clarity on effective approaches to prevention, it would be unwise to embark on another trial without thorough attention to the early phases described in the MRC framework.

The Birmingham healthy Eating and Active lifestyle for CHildren Study (BEACHeS, http://www.birmingham.ac.uk/research/activity/mds/projects/HaPS/PHEB/WAVES/BEACHeS/index.aspx), used the theoretical and modelling phases of the MRC framework to develop a multifaceted childhood obesity prevention programme, targeting SA children (phase I).[12] Here we report on the feasibility study (phase II). The aim was to assess feasibility and acceptability of the intervention. In addition we wanted to obtain data to inform a definitive (phase III) cluster randomised controlled trial (RCT).

## MATERIALS AND METHODS

The feasibility study was conducted in eight Birmingham primary schools from 2006 to 2009. Children underwent baseline measures between December 2006 and June 2007. Four schools were selected to receive the intervention (2007/2008 academic year), and the remainder had no active intervention. Follow-up data were collected 2 years after baseline.

### Setting

Birmingham is UK's second city with a high minority ethnic population (34%), one-fifth being from the three main SA communities (Pakistani, Bangladeshi and Indian). We obtained a list of all Local Authority-maintained primary schools in Birmingham. Of 304 schools, 52 had ≥50% of pupils from SA background (mean 75%). These, compared with the remainder, had a higher proportion of children eligible for free school meals (FSM), indicating higher deprivation. Schools were ranked in order of FSM eligibility, and those from either extreme were successively invited until eight agreed to take part.

### Participants

Pupils from years 1 and 2 (aged 5–7 years) were invited to participate. Parents of the children were approached by letter distributed through the schools, and active opt-in consent was sought for their child to participate in measurements. Consent for participation in the intervention was sought at the school level.

### Baseline and follow-up measures

Age, sex and ethnicity data (from parent report at school entry) were obtained from school records on all eligible children in participating schools. Children with consent also underwent a range of anthropometric measurements, including standing height (measured to nearest 0.1 cm with a Leicester Height Measure), weight (measured to nearest 0.1 kg with a Tanita bioimpedance monitor), two measures of waist circumference (measured to nearest 0.5 cm) and skinfold thickness at five sites (biceps, triceps, subscapular, suprailiac and thigh; measured using a Holtain calliper). Children also completed interviewer-administered questionnaires (not discussed in this paper, but including: quality of life (PedsQL),[13] self-concept (Marsh self-description questionnaire),[14] perceived physical competence (Harter Pictorial Scale for Young Children)[15] and body image perception (adapted Collin's Pictorial Image Scale)[16]). All measures were undertaken by trained researchers using standard protocols.

Dietary intake was assessed using the Child And Dietary Evaluation Tool (CADET)[17]; a 24 h food tick list that has been validated against a semiweighed diary in children aged 3–7 years. A researcher completed the CADET for children during school hours, and parents were given instructions for completing it for the remainder of the 24 h period. Physical activity levels were assessed using the Actiheart monitor (CamnTech, Papworth, UK) worn for five consecutive days, including a weekend. This is validated for use in children[18] and was set up to measure acceleration and heart rate at 30 s epochs. In addition, parents were asked to complete questionnaires which included questions on family composition, and family dietary and physical activity habits.

### Intervention

The process for intervention development has been reported elsewhere,[12] but in brief, the multicomponent intervention was developed by combining evidence from the literature with views from key stakeholders drawn from SA communities (including parents, teachers, school nurses, dieticians, community leaders, school governors and retail and leisure representatives close to

schools) and a multidisciplinary group of relevant professionals. Important contextual data were gained from stakeholders, which was critical for informing intervention development and highlighted potential barriers (eg, cultural unacceptability of certain types of physical activity for girls), as well as opportunities for intervention (eg, schools being considered a natural environment for providing skills to families), in relation to SA communities.[18 19] A review of local facilities, resources and opportunities related to healthy eating and the promotion of physical activity targeting children was used to inform the design and encourage longer term sustainability of the intervention. We also took account of national childhood obesity prevention policy during the development process to try and ensure that the intervention had an impact that was additional to existing national initiatives. The intervention targeted both diet and physical activity behaviours and consisted of two main strands: (1) increasing children's physical activity levels and promoting healthy eating through schools and (2) increasing skills among family members through family educational activities. A number of intervention techniques (as defined in the CALO-RE Taxonomy of behaviour-change techniques for physical activity and healthy eating[20]) were utilised to deliver each intervention component. A more detailed description of the intervention is provided in table 1.

### Allocation of intervention

This was a non-randomised feasibility trial. After baseline measurements were completed, schools were allocated to intervention or control arms. We matched schools by size, and proportion of children eligible for FSM. We then took the geographical location of the schools into account and allocated the matched pairs to either the intervention or control arm so that we minimised the chance of contamination between the two arms.

### Process measures

The main aim of the study was to assess intervention feasibility and acceptability. Each component was evaluated separately, using a variety of methods. These included collection of uptake data, direct observation, questionnaires to children and parents and interviews with key school staff. The questionnaires were also used to evaluate overall perceptions of the intervention and engagement with different intervention components. Topics covered in the semistructured interviews included exploration of how the different intervention components were implemented, which elements were perceived to work well and ideas for further development. The interviews were tape-recorded, transcribed and analysed thematically.

### Other measures and analysis

We assessed the feasibility of obtaining outcome data, primarily body mass index (BMI), and also diet and physical activity and other anthropometric measures as described above.

Exploratory comparison between intervention and control children was also undertaken to determine effect size.

Height and weight data were used to calculate BMI ($kg/m^2$) and converted into SD scores (BMI z-score) using the UK 1990 growth reference charts.[21] Children were categorised as underweight, healthy weight, overweight or obese using the 2nd, 85th and 95th centile cut-offs. For waist circumference and skinfolds, the mean was used for analyses. Skinfold measures were combined to obtain sum, upper (biceps, triceps and subscapular) and lower (suprailiac and thigh) skinfolds.

Data from the CADET were coded and analysed by a food diary analysis programme (DANTE, University of Leeds) to estimate total energy intake (kJ) and amount of fruit and vegetables, and sugar consumed. Data on foods consumed in school and at home were analysed separately, then combined to obtain estimates for the complete 24 h measurement period.

Accelerometry data were used to assess physical activity levels. Total daily volume of physical activity was estimated and expressed as average counts per minute (cpm). The mean duration of daily moderate-to-vigorous physical activity (MVPA, min/day) was calculated (400 cpm cut-off for lower threshold).[22] The proportion of children participating in ≥60 min MVPA (as recommended by international guidelines) was also calculated.

Statistical analyses were undertaken using STATA (V.11). The intraclass correlation coefficient (ICC) for the main outcome (BMI z-score) was calculated, but because of the small number of schools, clustering was not taken into account in the analysis. We analysed final BMI, diet and physical activity levels of children in the intervention, compared with those in the control group. To adjust for baseline differences, we initially developed multiple linear regression models, which included the relevant baseline values of BMI, dietary factors or physical activity measures as covariates. Further models were then developed which also included potential confounders as covariates (age, sex and ethnicity). Logistic regression was used to assess risk of obesity (compared with all non-obese children), and likelihood of meeting ≥60 min MVPA at follow-up in the intervention, compared with control children.

### RESULTS
### Feasibility and acceptability of intervention components

Some intervention components (particularly those delivered through school) were more successfully delivered than others. The intervention components were modified during the course of delivery to optimise participation and in response to feedback. The findings are summarised in table 1, and details are reported in the online supplementary appendix.

Two intervention components were found to be unsuitable to include within an intervention programme in the format delivered. First was the scheme to incentivise out of school leisure activities. Poor co-operation

**Table 1** Intervention components and techniques[20] included in the BEACHeS intervention programme and findings from the process evaluation

| Intervention component | Aim | Intervention techniques | Description | Agent responsible for delivery | Evaluation method | Evaluation findings |
|---|---|---|---|---|---|---|
| | | | **School-based activities** | | | |
| Physical activities within school day | To increase the amount of time that children are physically active within the school day | Environmental restructuring Prompt practice | Three elements introduced into schools 1. 'Wake Up Shake Up': a short (10 min) organised daily dance or exercise routine to music 2. Organised playground activities at lunch and break times through the training of school staff to act as 'play leaders' 3. 'Take 10': teaching resource which links 10 min physical activity in the classroom to curricular subjects | Trained school staff (including teachers, teaching assistants or lunch time assistants). The decision of which staff members to train for this component took into account individual school circumstances and was made in consultation with each school | ▶ Interviews with school staff ▶ Observation of sessions in schools ▶ Self-completion questionnaires administered to children and parents | Overall, school staff with a responsibility for health were enthusiastic and committed to introducing these schemes, and all schemes were acceptable to children. Individual school and staff factors strongly influenced the success of each element in the different schools. Parents, in general felt that the amount of physical activity their children were undertaking in school had increased over the last year |
| Incentive scheme to encourage physical activity out of school | To increase the amount of time outside of school hours that children spend doing leisure physical activities | Prompt self-monitoring of behaviour Prompt practice Provide rewards contingent on successful behaviour | Children received a sticker collection card from school and information on local participating sports and leisure venues. Each time a child attended a venue, they collected a sticker. The child with the most stickers in each school received a prize | Sticker collection card delivered through school class teacher. Stickers handed out by staff at leisure venues | ▶ Interviews with school staff ▶ Telephone survey of leisure venue staff ▶ Assessment of returned collection cards ▶ Questionnaires to children | Although this type of incentive scheme appears acceptable to children, parents and school staff alike, it was not feasible in terms of maintaining cooperation of participating venues. An element that was well received and could be retained, is the signposting information given to children and families |
| Attendance at a course run by a Premier league football club | To encourage physical activity and healthy eating through an iconic sporting institution | Provide information on consequences of behaviour Model/ demonstrate | School classes attend a 'Villa Vitality' day. Half the day is spent with Football Club coaches, exercising and learning football skills, and the | Aston Villa Football Club Community programme staff deliver on day of visit to club School class teachers | ▶ Interviews with school staff ▶ Self-completion questionnaires to parents | This was highly acceptable to children and school staff and is feasible to deliver to the target age group. There is some evidence that it |

**Table 1** Continued

| Intervention component | Aim | Intervention techniques | Description | Agent responsible for delivery | Evaluation method | Evaluation findings |
|---|---|---|---|---|---|---|
| | | behaviour Prompt identification of role model/ advocate Goal setting (behaviour) | other half of the day is an interactive learning session on healthy eating and healthy lifestyles. Teachers provided with material to deliver over 6 weeks to reinforce messages, and encourage weekly challenges or goals | deliver 6 weekly lessons after club visit | ▶ Preintervention and 6-week postintervention questionnaires (knowledge, attitudes and behaviour) administered to children | may favourably alter children's health-behaviours |
| | | | **Increasing skills of families through activity-based learning** | | | |
| Cooking courses for family members | To increase healthy cooking skills, and confidenced and influence dietary behaviour | Provide information on consequences of behaviour in general Model/ demonstrate behaviour Provide instruction on how to perform the behaviour Prompt generalisation of behaviour | Five-week courses on healthy cooking were delivered through schools to parents or other family members, some courses include children. Courses ran successively to allow all parents to attend if they wanted. Healthy recipes were distributed to support the course content | Birmingham Community NHS Trust dietetics staff | ▶ Interviews with school staff ▶ Uptake rates for courses ▶ Participant precourse and postcourse questionnaires | This component was popular with those who participated and there was some evidence that it influenced confidence and cooking practices. Running sessions for parents and children was the most popular model, and having the sessions based in school time for children and inviting parents to attend improved attendance |
| Information on local leisure opportunities and 'taster' sessions for families | To equip families with the knowledge and skills to undertake physical activities with their children in their leisure time | Provide information on when and where to perform behaviour Model/ demonstrate behaviour | Parents were given information on local sporting and leisure venues and events. They were invited to attend weekend taster sessions with their children, through school. Activities ranged, from cricket and football, to archery, climbing and dry-slope skiing. There was no cost for the activities and transport was provided | BEACHeS research staff compiled lists of venues and prepared signposting sheets. BEACHeS research staff accompanied families to leisure venues, where leisure venue staff delivered sessions | ▶ Interviews with school staff ▶ Uptake of the taster sessions ▶ Self-completion questionnaires to parents and children | This component was resource intensive to deliver, and uptake was very low. However, the signposting information was used by families, and was appreciated |

**Table 1** Continued

| Intervention component | Aim | Intervention techniques | Description | Agent responsible for delivery | Evaluation method | Evaluation findings |
|---|---|---|---|---|---|---|
| Training walk leaders to initiate community walking programmes | To increase walking by families and other community members through organised leisure walks | Model/demonstrate behaviour; Prompt practice | Community volunteers were recruited through schools to become trained walk leaders. Training was provided to equip volunteers to organise and lead walks in their local community | Walk-leader training programme delivered through Heart of Birmingham NHS Trust | ▸ Assessment of numbers attending training to be walk leaders ▸ Monitoring numbers joining walking groups | This component proved unfeasible as there was a lack of volunteers. Even those who expressed an initial interest failed to attend the training. Despite repeated efforts to recruit community volunteers, no one attended the training in any of the four BEACHeS intervention communities |

BEACHeS, The Birmingham healthy Eating and Active lifestyle for CHildren Study.

from leisure venues and lack of resources to continually remind and motivate children contributed to the failure of this component. Second was the training of walk leaders. Despite efforts to recruit through school staff, influential parents and various forms of publicity, volunteers were not forthcoming. The only person who underwent training did not undertake any walking groups.

One component was partially successful. Signposting of leisure facilities in the local area was popular among parents and school staff. However, attendance at organised taster sessions was poor, which was outweighed by the high staff and monetary resources required to deliver the component. The taster sessions were therefore not feasible to include in a larger trial.

The other intervention components had varying degrees of success, and the process evaluation highlighted how delivery could be improved. Individual school characteristics and differences between staff members strongly influenced the success of each element in the different schools.

### Acceptability of allocation to control

Acceptability of non-intervention was assessed through interviews with control school staff. All understood the need for a control arm. One would have liked alternative support to compensate for not being offered the intervention. In other control schools, staff expressed that being part of the study had benefitted them in other ways, and contributed to the school's status as a 'healthy school'.

### Outcomes
#### Feasibility of outcome measures
There were 1090 eligible children in the eight participating schools (range 54–180). Of these, 606 (55.6%) had parental consent and anthropometric measures were completed on those in school on measurement days (n=571, 94.2% of consented). Useable data (≥3 days) for Actiheart were available for 508 (89%). Completion of CADET was more variable. Although 445 (77.9%) were returned at baseline, 269 (47.1%) were complete, of which two-thirds (n=174) had usable data. Two years after the baseline measures, 488 children (85.5%) were successfully followed up. The proportion with usable Actiheart data was similar to baseline. However a higher proportion (n=454, 93%) had a completed CADET, although only 163 (36%) had usable home data.

#### Findings from the feasibility study
A total of 574 children were included in the trial (figure 1), of whom 85.9% were SA. Baseline characteristics are summarised in table 2 (anthropometric measures were completed for 571 of the 574 participating children). The age, sex and ethnicity of those who took part were similar in distribution to the characteristics of the non-consented eligible children.

Over 90% in both arms were from the most deprived areas, in keeping with the location and ethnic composition of the population. Around one in five (n=115) were overweight or obese. This proportion was slightly higher in the control (21.7%) compared to the intervention (18.3%) schools. A similar pattern was seen for other measures of body fat (skinfold measures, bioimpedance), but not for waist circumference which was similar in intervention and control groups. Under half of the children (47.9%) undertook ≥60 min of MVPA. Levels of physical activity (total cpm) and duration of time spent in MVPA were slightly higher among children from control, compared with intervention schools. Total dietary energy, fruit and vegetable and sugar intake were slightly higher among children in intervention, compared with control schools.

Two years postbaseline, 254 (83.3%) children in the control and 234 (86.2%) in the intervention schools were successfully followed up. There was no significant difference in baseline weight status, MVPA, diet or sex, between those followed up and those lost to follow-up (data not shown). However, SA children were less likely to be lost to follow-up (n=58; 11.9%) compared with those from other ethnic groups (n=28; 34.6%).

## Estimation of effect size to inform definitive trial

Anthropometric, physical activity and dietary measures in intervention and control groups at follow-up are shown in table 3. The proportion of children who were overweight or obese had increased in all schools from baseline to follow-up (from 7.3% to 9.9%, and from 12.8% to 19.1% for overweight and obese, respectively). The risk of obesity was significantly lower in the intervention compared with the control group (OR 0.41; 95% CI 0.19 to 0.89). The increase in BMI z-score was also significantly lower in the intervention compared with control, after adjustment ($-0.15\,\mathrm{kg/m^2}$; 95% CI $-0.27$ to $-0.03$; table 4). A similar trend was seen for all other anthropometric measures, although none were statistically significant.

The ICC for the outcome 'overweight/obese' compared to non-overweight, was 0.00 (95% CI 0 to 0.02), while for BMI z-score, the ICC was 0.01 (95% CI 0 to 0.04). Therefore, taking account of clustering in the analysis would make marginal difference to the findings.

**Figure 1** Flow diagram of recruitment and follow-up of participants in the feasibility study.

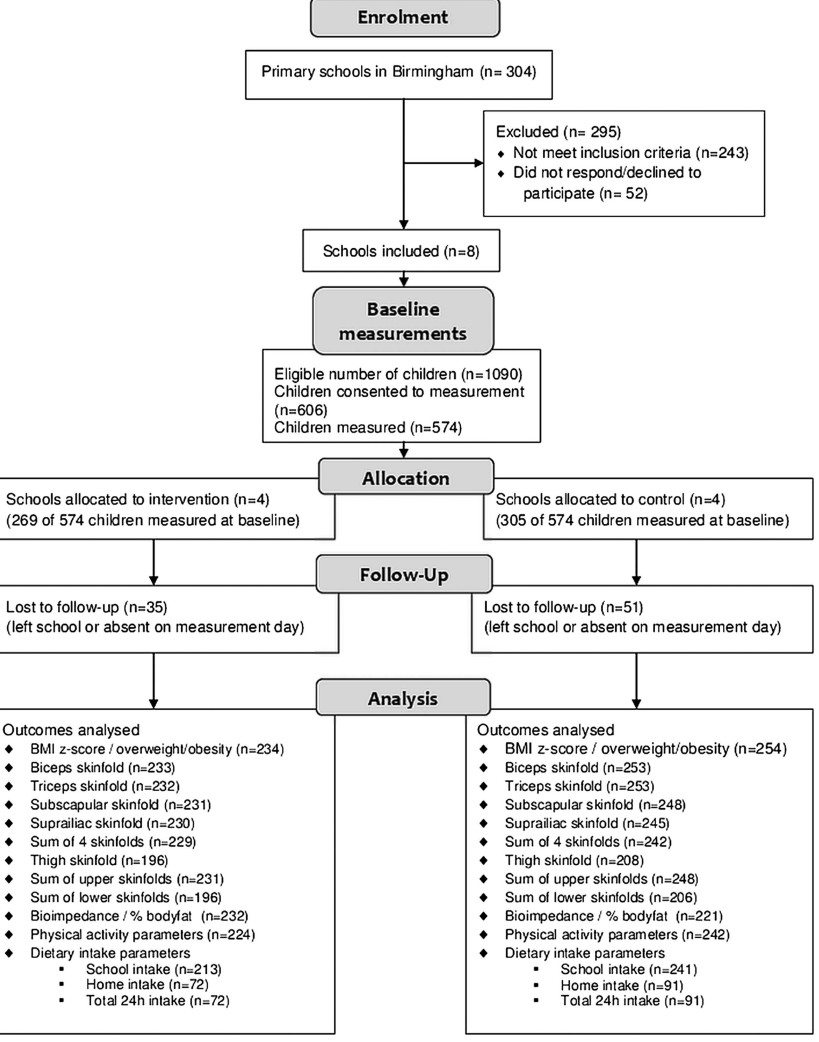

**Table 2** Baseline characteristics of children measured for the BEACHeS feasibility study

| Characteristic/measure* | Intervention group: N=269 n (%) or mean (SD) | Control group: N=305 n (%) or mean (SD) |
|---|---|---|
| Sex (n=574) | | |
| Male | 144 (53.5) | 142 (49.8) |
| Female | 125 (46.5) | 153 (50.2) |
| Age in years (n=574) | 6.53 (0.59) | 6.44 (0.58) |
| Ethnicity (n=574) | | |
| Bangladeshi | 36 (13.4) | 46 (15.1) |
| Indian | 22 (8.2) | 5 (1.6) |
| Pakistani | 181 (67.3) | 203 (66.6) |
| Other | 30 (11.2) | 51 (16.7) |
| Townsend score decile (n=572) | | |
| 1 (most deprived) | 250 (93.3) | 285 (93.8) |
| 2 | 6 (2.2) | 9 (3.0) |
| 3 | 8 (3.0) | 5 (1.6) |
| 4–7 | 4 (1.5) | 5 (1.6) |
| BMI-SDS score (n=571) | -0.03 (1.37) | 0.08 (1.39) |
| Weight status (n=571) | | |
| Underweight | 6 (2.3) | 10 (3.3) |
| Healthy weight | 212 (79.4) | 228 (75.0) |
| Overweight | 15 (5.6) | 27 (8.9) |
| Obese | 34 (12.7) | 39 (12.8) |
| Waist circumference (cm) (n=569) | 55.6 (7.7) | 55.3 (6.9) |
| Skinfold measures (mm) | | |
| Biceps (n=563) | 7.5 (3.6) | 8.0 (4.0) |
| Triceps (n=563) | 10.9 (4.1) | 11.6 (4.5) |
| Subscapular (n=559) | 7.5 (4.4) | 7.9 (5.0) |
| Suprailiac (n=561) | 7.0 (4.4) | 7.4 (4.7) |
| Sum of 4 skinfolds (n=556) | 32.5 (14.7) | 34.8 (16.8) |
| Thigh (n=433) | 14.4 (5.5) | 15.7 (6.3) |
| Sum of upper skinfolds (n=557) | 25.7 (10.9) | 27.5 (12.5) |
| Sum of lower skinfolds (n=433) | 21.0 (8.8) | 22.9 (10.1) |
| Bioimpedance ($\Omega$) (n=521) | 692.6 (72.5) | 695.1 (80.8) |
| Physical activity (n=535) | | |
| Counts/min | 79.9 (23.4) | 83.4 (27.3) |
| MVPA min/24 h | 52.8 (28.4) | 62.9 (25.0) |
| ≥60 mins MVPA per day (n=535) | 100 (40.2) | 156 (54.5) |
| 1 day school dietary intake (n=441) | | |
| Mean energy (kJ) | 2378.2 (1619.2) | 1917.1 (1821.7) |
| Fruit and vegetables (g) | 140.6 (121.4) | 105.8 (118.7) |
| Sugar (g) | 41.0 (60.9) | 35.3 (62.9) |
| 1 day home dietary intake (n=174) | | |
| Mean energy (kJ) | 7021.3 (3181.1) | 6058.34 (3138.2) |
| Fruit and vegetables (g) | 329.85 (232.2) | 267.95 (193.3) |
| Sugar (g) | 105.36 (50.2) | 93.50 (36.6) |
| 24 h dietary intake (n=173) | | |
| Total energy (kJ) | 9326.6 (3083.7) | 8397.3 (4035.5) |
| Fruit and vegetables (g) | 475.6 (261.4) | 368.7 (220.2) |
| Sugar (g) | 154.1 (108.7) | 129.7 (85.8) |

*n in this column indicates how many children had useable data for each characteristic/measure.
BEACHeS, The Birmingham healthy Eating and Active lifestyle for CHildren Study; BMI, body mass index; MVPA, moderate-to-vigorous physical activity.

The proportion of children who undertook ≥60 min MVPA reduced (from 48.8% to 27.1%) at follow-up, with the reduction being greater among control (30.2%) compared with intervention (23.8%) children. The differences in physical activity levels at follow-up were not significant between groups (table 4).

Total energy intake had increased slightly at follow-up (7473 kJ at baseline to 8130 kJ at follow-up). There were no significant differences in dietary intake between control and intervention children, although 24 h dietary intake data were only available for 163 (33%) children at follow-up, and only 61 children had dietary data at both baseline and follow-up.

**Table 3** Anthropometric, diet and physical activity measures in intervention and control groups at follow-up

| Measure* | Intervention group (N=234) n (%) or mean (SD) | Control group (N=254) n (%) or mean (SD) |
|---|---|---|
| Anthropometric measures | | |
| BMI z-score (n=488) | 0.13 (1.5) | 0.40 (1.5) |
| Weight status (n=488) | | |
| Underweight | 11 (4.7) | 8 (3.2) |
| Healthy weight | 160 (68.4) | 168 (66.1) |
| Overweight | 27 (11.5) | 21 (8.3) |
| Obese | 36 (15.4) | 57 (22.4) |
| Waist circumference (cm) (n=472) | 59.4 (9.5) | 60.4 (9.1) |
| Skinfold measures (mm) | | |
| Biceps (n=486) | 6.9 (3.5) | 7.7 (3.8) |
| Triceps (n=485) | 11.2 (4.8) | 11.9 (4.6) |
| Subscapular (n=479) | 8.5 (5.3) | 9.3 (5.8) |
| Suprailiac (n=475) | 8.8 (5.9) | 9.4 (5.9) |
| Sum of 4 skinfolds (n=471) | 35.2 (18.1) | 37.6 (18.4) |
| Thigh (n=404) | 17.3 (7.5) | 18.9 (8.1) |
| Sum of upper skinfolds (n=479) | 26.5 (12.7) | 28.7 (13.1) |
| Sum of lower skinfolds (n=402) | 25.3 (11.8) | 27.6 (13.1) |
| Bioimpedance ($\Omega$) (n=453) | 692.0 (83.1) | 688.3 (81.3) |
| Physical activity (n=467) | | |
| Counts/min | 68.7 (33.4) | 71.0 (22.9) |
| MVPA min/24 h | 49.1 (21.8) | 51.1 (20.2) |
| Achieving ≥60 min MVPA | 53 (23.6) | 73 (30.2) |
| Dietary intake | | |
| School (n=454) | | |
| Energy (kJ) | 1908.7 (831.8) | 2045.1 (777.9) |
| Fruit and vegetables (g) | 143.1 (135.0) | 93.9 (94.0) |
| Sugar (g) | 25.0 (15.7) | 29.8 (16.7) |
| Home (n=163) | | |
| Energy (kJ) | 7860.5 (4366.4) | 8145.4 (4004.5) |
| Fruit and vegetables (g) | 367.5 (316.6) | 342.0 (224.9) |
| Sugar (g) | 113.8 (63.7) | 121.0 (56.8) |
| 24 h dietary intake (n=163) | | |
| Energy (kJ) | 9527.8 (4400.3) | 9820.7 (3773.1) |
| Fruit and vegetables (g) | 519.1 (350.2) | 446.0 (238.5) |
| Sugar (g) | 137.2 (64.2) | 150.5 (59.9) |

*n in this column indicates how many children had useable data for each measure.
BMI, body mass index; MVPA, moderate-to-vigorous physical activity.

However, school dietary intake data were more complete (93% with follow-up data and 73% with both baseline and follow-up data), and children in intervention schools had significantly more fruits and vegetables and lower sugar intake, compared with those in control schools (table 4).

As the intervention was designed to be particularly relevant to SA children, we repeated the multivariate analyses including only children of SA ethnicity. The mean differences and ORs for the outcomes were of a similar magnitude to the main analyses (data not shown).

## DISCUSSION

We demonstrated the feasibility of delivering a multicomponent obesity prevention intervention targeting dietary and physical activity behaviours to a socioeconomically disadvantaged, multiethnic population of primary school-aged children. The feasibility study provided an opportunity to refine and modify the programme and yielded important information on acceptability and feasibility of the intervention and measurements required for assessing outcomes in a definitive RCT.

### Strengths and limitations

This is one of few studies focusing on SA populations, which comprise the largest minority ethnic group in the UK, with higher risk of obesity and its consequences. The iterative process of intervention refinement was informed by the MRC framework for complex interventions. While the framework has been used for the development of other interventions in National Health Service (NHS) settings, we have demonstrated its use in the wider community setting.

The components of the intervention were influenced by stakeholder views and available resources, thus its applicability for wider populations and settings is potentially

**Table 4** Adjusted differences in anthropometric, diet and physical activity measures between control and intervention groups

| Outcome variable* | Intervention vs control (adjusted for baseline) OR (95% CI) | p Value | Intervention vs control (finally adjusted)† Adjusted†OR (95% CI) | p Value |
|---|---|---|---|---|
| Obese (n=486) | 0.36 (0.17 to 0.77) | 0.01 | 0.41 (0.19 to 0.89) | 0.02 |
| Achieving ≥60 min MVPA (n=441) | 0.82 (0.52 to 1.28) | 0.38 | 0.74 (0.45 to 1.20) | 0.22 |
| | **Mean difference (95% CI)** | | **Adjusted† mean difference (95% CI)** | |
| BMI z-score (n=486) | −0.15 (−0.26 to −0.03) | 0.02 | −0.15 (−0.27 to −0.03) | 0.02 |
| Waist circumference (cm) (n=482) | −0.88 (−1.87 to 0.10) | 0.08 | −0.86 (−1.87 to 0.15) | 0.09 |
| Skinfold measures (mm) | | | | |
| Biceps (n=479) | −0.48 (−0.98 to 0.01) | 0.06 | −0.44 (−0.93 to 0.06) | 0.08 |
| Triceps (n=478) | −0.14 (−0.68 to 0.40) | 0.61 | −0.10 (−0.64 to 0.45) | 0.71 |
| Subscapular (n=469) | −0.46 (−0.98 to 0.06) | 0.09 | −0.38 (−0.89 to 0.14) | 0.15 |
| Suprailiac (n=468) | −0.23 (−0.84 to 0.37) | 0.45 | −0.23 (−0.83 to 0.37) | 0.46 |
| Sum of 4 skinfolds (n=461) | −1.09 (−2.85 to 0.67) | 0.23 | −0.97 (−2.70 to 0.77) | 0.27 |
| Thigh (n=324) | −0.31 (−1.39 to 0.78) | 0.58 | −0.27 (−1.38 to 0.84) | 0.63 |
| Sum of upper skinfolds (n=468) | −0.90 (−2.21 to 0.42) | 0.18 | −0.76 (−2.05 to 0.53) | 0.25 |
| Sum of lower skinfolds (n=323) | −0.36 (−1.91 to 1.19) | 0.65 | −0.40 (−1.98 to 1.18) | 0.62 |
| Bioimpedance (Ω) (n=409) | 3.33 (−5.23 to 11.89) | 0.45 | 3.50 (−5.14 to 12.15) | 0.43 |
| Counts/min (increments of 20) (n=441) | −0.15 (−0.34 to 0.04) | 0.12 | −0.18 (−0.36 to 0.01) | 0.06 |
| MVPA min/24 h (n=441) | 1.52 (−2.14 to 5.17) | 0.42 | 0.51 (−2.97 to 3.99) | 0.77 |
| School | | | | |
| Energy (kJ) (n=358) | −78.78 (−240.75 to 83.18) | 0.34 | −86.02 (−250.29 to 78.20) | 0.30 |
| Fruit and vegetables (g) (n=358) | 59.88 (34.56 to 85.19) | <0.001 | 63.35 (37.53 to 89.17) | <0.001 |
| Sugar (g) (n=358) | −3.86 (−7.27 to −0.45) | 0.03 | −3.86 (−7.37 to −0.36) | 0.03 |
| Home | | | | |
| Energy (kJ) (n=61) | 1322.94 (−292.75 to 2938.60) | 0.12 | 1534.73 (−117.74 to 3187.20) | 0.07 |
| Fruit and vegetables (g) (n=61) | 21.61 (−81.26 to 124.47) | 0.68 | 18.98 (−89.43 to 127.40) | 0.73 |
| Sugar (g) (n=61) | 6.88 (−17.04 to 30.81) | 0.57 | 9.17 (−15.16 to 33.51) | 0.45 |
| 24 h dietary intake | | | | |
| Energy (kJ) (n=61) | 883.16 (−888.31 to 2654.66) | 0.32 | 1092.36 (−723.33 to 2908.09) | 0.23 |
| Fruit and vegetables (g) (n=61) | 89.64 (−32.51 to 211.79) | 0.15 | 86.70 (−42.92 to 216.32) | 0.19 |

Continued

limited. However, the multifaceted intervention aimed to modify school and family environments and included elements that have been identified as promising in systematic reviews.[7] [23] Furthermore the intervention components have theoretical validity for behaviour change in any population, and the incorporated techniques are transferrable. The targeting of SA stakeholders for intervention development is likely to have allowed us to exclude intervention components that would not be acceptable to this subpopulation. Nevertheless, the developed intervention is likely to be acceptable not only in these ethnic groups, but also in the wider UK population.

Delivery of intervention, undertaken by staff outside the research team, was non-standardised. This allowed a pragmatic approach to be tested, which could be more easily rolled out. Intervention components delivered directly to the children and through school staff (physical activity component and Villa Vitality) were more likely to have high uptake than those delivered to families (leisure taster sessions or walk leader training). The complexity of delivering community-based interventions targeting children probably explains why most previous trials are school based.

During the trial, all children in schools allocated to the intervention arm were exposed to the intervention components. However, only about half had consent for measurements. We found no significant differences in sex and ethnicity between consented and non-consented children. Further, the distribution of weight status among children who were measured is similar to national data for this age group,[24] suggesting that selection bias was unlikely.

## Intervention acceptability and feasibility

A variety of intervention techniques were incorporated with variable success. Environmental restructuring (structured physical activity and play opportunities in school) was feasible and generally accepted. Demonstration of the target behaviour and prompting practice (Villa Vitality, cooking workshops, taster activity sessions and walking groups) had mixed results. Apart from Villa Vitality which was incorporated within the school setting, there was limited participation, despite enthusiasm among those who did take part. At a population level, these types of intervention are less feasible to deliver, unless they are incorporated within the school setting. Providing information and prompting identification of role models were feasible and acceptable and would be replicable in a larger trial. Techniques to prompt self-monitoring and rewarding successful behaviour were acceptable, but had limited success in this community setting.

During the period of intervention delivery, we used a variety of methods and involved different stakeholders (school staff, parents and children), to assess the acceptability of the intervention components. We also allowed the programme to be modified and the implementation of elements to vary in the different intervention schools. This tailoring to the local school context was critical in determining the success of the intervention. For example in one school, lunchtime supervisors were trained to deliver a structured physical activity programme at lunchtime, but did not go on to deliver the programme. Following this failure of implementation, an enthusiastic teaching assistant was trained, who successfully delivered the intervention. Thus, while standardisation of aspects of the intervention is important, some scope for tailoring to local context in terms of implementation and delivery needs to be considered.[25]

## Informing a definitive RCT

The intervention was aimed at predominantly SA populations residing in inner city settings. Despite challenges, including language barriers, 80% were successfully followed up. We demonstrated the feasibility of undertaking a wide range of anthropometric measures within school and the feasibility of Actiheart monitors for assessing physical activity in free living children (approximately 90% had usable data). Assessment of dietary intake was less successful at baseline, mainly due to language barriers and difficulties for parents in completing the forms, but the feasibility study allowed us to refine the administration of the tool, so that measurement was more complete at follow-up.

Although the feasibility study was not powered to examine intervention outcomes, we did find that the direction of effect for most outcomes were in favour of the intervention, supporting the need for a definitive trial. In particular, at follow-up children in intervention schools had BMI z-scores on average $0.15 \text{ kg/m}^2$ lower than children in control schools, which is in keeping with the effect size reported in a meta-analysis of childhood obesity prevention trials.[7]

The costs of the intervention were not formally examined, as this was a feasibility study and the intervention components were being modified and tested. Nevertheless the feasibility stage provided an opportunity to consider resource requirements and to modify the intervention accordingly to inform a definitive study. In order to ensure sustainability, most intervention components were adapted from existing services commissioned by the local NHS bodies at the time (including Villa Vitality, cooking courses and training of walk leaders). The resources for training teachers to deliver structured physical activity sessions are available commercially to schools, and were compiled by the research team. The signposting information for local leisure facilities and for the weekend activities was similarly compiled by the research team, summarising already available services and facilities.

## Conclusions

We have used the MRC framework for complex interventions to develop a childhood obesity prevention intervention that can be evaluated within the context of a cluster RCT. Although the intervention was informed by stakeholders, and evidence and guidelines from previous literature, some elements were found not to be feasible or acceptable to participants in practice. The feasibility study was an essential step in finalising the intervention programme prior to definitive evaluation. Based on the findings from this study, a definitive cluster RCT is currently underway to assess the clinical and cost-effectiveness of the finalised intervention in primary school children (ISRCTN97000586).

**Author affiliations**
[1]School of Health & Population Sciences (Public Health, Epidemiology and Biostatistics and Primary Care Clinical Sciences), University of Birmingham, Birmingham, UK
[2]Nutritional Epidemiology Group, School of Food Science and Nutrition, University of Leeds, Leeds, UK
[3]MRC Epidemiology Unit, Institute of Metabolic Science, Cambridge, UK
[4]Department of Sport Medicine, Norwegian School of Sport Sciences, Cambridge, UK
[5]School of Clinical and Experimental Medicine, University of Birmingham, Birmingham, UK
[6]School of Sport, Exercise and Rehabilitation Sciences, The University of Birmingham, Birmingham, UK
[7]Edinburgh Ethnicity and Health Research Group, Centre for Population Health Sciences, University of Edinburgh, Edinburgh, UK

**Acknowledgements** S Passmore (Birmingham City Council) advised the study team on identification and engaging of schools and pupils, and the tailoring of interventions to be acceptable to schools. M Howard (Heart of Birmingham PCT) facilitated the delivery of some of the intervention components through the Primary Care Trust. E McGee (Birmingham Community Nutrition and Dietetic Service) oversaw the delivery of the cooking workshops and contributed to their evaluation. K Westgate and S Mayle (MRC Epidemiology Unit, Cambridge) undertook cleaning, reducing and analysing the physical activity data. C Cleghorn (University of Leeds) contributed to the development of the CADET, oversaw staff training and administration of the tool, and undertook cleaning and analysis of the dietary data. The authors are grateful to all the funding partners for their support: British Heart Foundation;

Cancer Research UK; Department of Health; Diabetes UK; Economic and Social Research Council; Medical Research Council; Research and Development Office for the Northern Ireland Health and Social Services; Chief Scientist Office, Scottish Executive Health Department; Welsh Assembly Government and World Cancer Research Fund.

**Contributors** PA conceptualised and designed the study overall, oversaw study planning, delivery and evaluation, wrote the analysis plan and drafted and revised the paper and approved the final manuscript as submitted. She is the guarantor. MP assisted in the overall delivery of the study, designed the process evaluation, and undertook the data cleaning and analysis. She has approved the final manuscript as submitted, and is also the guarantor. JC designed the dietary data assessment tool, oversaw training for researchers collecting data and the analysis of the dietary data obtained. She has approved the final manuscript as submitted. UE advised on physical activity assessment, provided training of researchers in collecting Actiheart data and oversaw the analysis of physical activity data from Actihearts. He has approved the final manuscript as submitted. TB advised on anthropomentric measurement tools used, arranged for training of research staff to undertake measures and advised on interpretation of the anthropometric data obtained. He has approved the final manuscript as submitted. AD advised on physical activity components of the intervention and use of incentives to motivate children. She has approved the final manuscript as submitted. JDe provided statistical support, advised on statistical analysis and on using the data obtained to inform sample size estimation for the definitive trial. He has approved the final manuscript as submitted. JDu advised on the psychological measurement instruments, advised research staff on child protection issues and contributed to shaping the physical activity components of the intervention. She has approved the final manuscript as submitted. PG advised on anthropometric measurements to be included, the sampling strategy and on ethnic minority health. He has approved the final manuscript as submitted. JP advised on process evaluation and contributed to the interpretation of qualitative data obtained. She has approved the final manuscript as submitted. RB advised on the anthropometric measurements, definition of the target population, tailoring of the intervention to be culturally appropriate and relevant literature on ethnicity and health. He also provided important comments on the final draft and has approved the final manuscript as submitted. KKC conceived the original idea informing this study, contributed to the planning and delivery. He has approved the final manuscript as submitted. All authors have contributed to the design of the intervention, advised on study progress and critically revised and approved the final manuscript.

**Funding** The Birmingham healthy Eating, Active lifestyle for Children Study (BEACHeS) was funded by the National Prevention Research Initiative (NPRI, http://www.mrc.ac.uk/Ourresearch/ResearchInitiatives/NPRI/index.htm), Grant no. G0501292

**Competing interests** None.

**Ethics approval** Ethical approval was granted by The Black Country Research Ethics Committee (08/H1202/22).

**Provenance and peer review** Not commissioned; externally peer reviewed.

**Data sharing statement** The BEACHeS study dataset is available on request from the study investigators.

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
