## [Reviewer comments · BMJ Open]

Some articles will have been accepted based in part or entirely on reviews undertaken for other BMJ Group journals. These will be reproduced where possible.

ARTICLE DETAILS

TITLE (PROVISIONAL)	Preventing childhood obesity, phase II feasibility study focusing on South Asians: BEACHeS
AUTHORS	Adab, Peymane; Pallan, Miranda; Cade, Janet; Ekelund, Ulf; Barrett, Timothy; Daley, Amanda; Deeks, Jon; Duda, Joan; Gill, Paramjit; Parry, Jayne; Bhopal, Raj; Cheng, Kar Keung

VERSION 1 - REVIEW

REVIEWER	Carolyn Summerbell Durham University, UK
REVIEW RETURNED	09-Feb-2014

GENERAL COMMENTS	This paper reports on an exploratory trial which aims to prevent childhood obesity in children who are 6-8 years of age. The intervention is delivered via schools where children are predominately South Asian. A number of the authors of the paper have significant expertise in this type of study, and topic area. The paper is well written, and easy to read. I have a few comments which are not fatal flaws, but may limit the study in terms of its importance: 1. It is unclear from the paper how the components of the intervention were included / adapted from previous interventions, in terms of being culturally appropriate. This, to me, is the most interesting aspect of the study, and there is little mention of this in the results and discussion. For example, if another researcher wanted to try out the intervention for a different cultural group, which components of the BEACHeS intervention are tailored specifically for South Asians, and why did some appear to be more feasible than others.2. It is not always obvious from the text who delivered the interventions. It would be useful if this was clear.3. There is no mention of costs of the intervention components (and this in part relates to point 2), which obviously need to be considered when thinking about sustainability and roll out. It is most curious that the authors make no mention of cost. It would be no great surprise if a more intense and expensive intervention, that may not be sustainable without significant external support and co-ordination appeared to be more effective than nothing. I appreciate that this is an exploratory study, but the authors should include information about costs in this paper.
--

	4. It appears from the results that the school based components of the intervention were more effective than those delivered external to the school. This is not surprising, but more discussion on this point would be useful. 5. 'calorific' and kcal should be changed to 'energy' and KJ throughout. 6. In terms of design, it is unclear how the researchers allocated school to control of intervention. This is so important, and I am surprised it is missing from the information in the paper. 7. Looking at Fig 1, it appears that baseline measurements of the children were taken after schools were allocated to intervention or control? 8. I am left wondering about various design aspects of this study, e.g. opt in/ opt out consent. It is such a shame that the authors did not publish the study protocol.
--	---

REVIEWER	Dr Victoria Allgar Hull and York Medical School University of York England
REVIEW RETURNED	25-Feb-2014

GENERAL COMMENTS	The analysis section is rather brief when discussing which statistical tests were undertaken. There is no detail about which software package was used. 81 children were described as 'other ethnicity'. If the focus is on SA children, I wonder if these should be excluded. The statement in the results section "Around one in five (n=115) were overweight or obese. This proportion was slightly higher in the control (21.7%) compared to the intervention (18.3%) schools, mainly due to a higher prevalence of overweight", is a little self explanatory. In table 2 there is missing data in some of the groups e.g. weight status for intervention is missing for 2 children - the total in each category should be made clear. For the analysis focusing on 2 year follow-up. It states that "At follow up, the proportion of children who were overweight or obese had increased in all schools (from 7.3% to 9.9%, and from 12.8% to 19.1% for overweight and obese respectively)." It would be interesting to see here the proportion of children at follow-up, who were obese at baseline and follow-up, as there is some missing data. Also what proportion are underweight, normal, obese/overweight, rather than just obese as shown in table 3. And the differences between the intervention and control groups. For the continuous data in table 3, it wasn't clear what technique had been used here - was it ANCOVA to adjust for baseline values? It would be interesting to see the mean differences between the intervention and control group, rather than odd's ratios.
---

In table 3 it would also be useful to see the numbers in each analysis to show the extent of missing data.
--

VERSION 1 – AUTHOR RESPONSE

Reviewer Name Carolyn Summerbell

We thank the reviewer for the generally positive comments about the paper and how it was written, and below outline how we have addressed the limitations she had highlighted:

1. It is unclear from the paper how the components of the intervention were included / adapted from previous interventions, in terms of being culturally appropriate. This, to me, is the most interesting aspect of the study, and there is little mention of this in the results and discussion. For example, if another researcher wanted to try out the intervention for a different cultural group, which components of the BEACHeS intervention are tailored specifically for South Asians, and why did some appear to be more feasible than others.

We had reported the process of intervention development and how the various components were decided on briefly on page 7, where we had also referred to two other publications which explain the process in much more detail. However, we have added some further detail to indicate the point about how we considered cultural adaptation on page 7/8. Furthermore, whilst the focus of intervention was SA communities, the intervention was being delivered to a multicultural population of school children. Thus the aim was not to tailor the intervention to a particular cultural group, but rather to ensure that it would be inclusive and consider cultural barriers, making it relevant to a multi-ethnic population. This point is already discussed on page 24, paragraph 3. It is also not possible to deduce whether the lack of acceptability of certain components were culture specific, as this was not an a-priori research question, and there was not sufficient representation from different ethnic groups for any meaningful comparisons.

2. It is not always obvious from the text who delivered the interventions. It would be useful if this was clear.

We have added a column to table 1 to clarify the agent responsible for delivery of each intervention component.

3. There is no mention of costs of the intervention components (and this in part relates to point 2), which obviously need to be considered when thinking about sustainability and roll out. It is most curious that the authors make no mention of cost. It would be no great surprise if a more intense and expensive intervention, that may not be sustainable without significant external support and co-ordination appeared to be more effective than nothing. I appreciate that this is an exploratory study, but the authors should include information about costs in this paper.

We did not formally evaluate costs, as assessment of cost-effectiveness is the remit of a definitive trial. However, we agree with the reviewer that there is a role for considering costs within a feasibility study, and we had referred to resource use within paragraphs 2&3 of the results section. Consideration of resource use helped to shape and modify the intervention. As this was not explicitly stated, we have now added a paragraph at the end of the discussion, on page 26 (under "Informing a definitive RCT" to say:

"The costs of the intervention were not formally examined, as this was a feasibility study and the intervention components were being modified and tested. Nevertheless the feasibility stage provided an opportunity to consider resource requirements and to modify the intervention accordingly to inform

a definitive study. In order to ensure sustainability, most intervention components were adapted from existing services commissioned by the local NHS bodies at the time (including Villa Vitality, cooking courses and training of walk leaders). The resources for training teachers to deliver structured physical activity sessions are available commercially to schools, and were compiled by the research team. The signposting information for local leisure facilities and for the weekend activities was similarly compiled by the research team, summarising already available services and facilities.”

4. It appears from the results that the school based components of the intervention were more effective than those delivered external to the school. This is not surprising, but more discussion on this point would be useful.

We have added two sentences to the discussion (pages 24/25) to highlight this point:

“Intervention components delivered directly to the children and through school staff (physical activity component and Villa Vitality) were more likely to have high uptake than those delivered to families (leisure taster sessions or walk leader training). The complexity of delivering community based interventions targeting children probably explains why most previous trials are school based.”

5. ‘caloric’ and kcal should be changed to ‘energy’ and KJ throughout.

We have now changed this in line with the suggestion.

6. In terms of design, it is unclear how the researchers allocated school to control of intervention. This is so important, and I am surprised it is missing from the information in the paper.

We did have a section on “Allocation of intervention” on page 12. However, we have now modified this section to make it even clearer how this was done.

7. Looking at Fig 1, it appears that baseline measurements of the children were taken after schools were allocated to intervention or control?

We agree that the figure is misleading. The allocation was in fact done after the baseline measures were undertaken. We have now modified figure 1 to clarify this.

8. I am left wondering about various design aspects of this study, e.g. opt in/ opt out consent. It is such a shame that the authors did not publish the study protocol.

Study measures were done using active opt-in consent. This is stated on page 6 of the methods, under “Participants”. We have added “opt-in” to the previous version where we stated “active” to make this clear.

Reviewer Name Dr Victoria Allgar

The analysis section is rather brief when discussing which statistical tests were undertaken. There is no detail about which software package was used.

The main focus of the feasibility study was to assess feasibility and acceptability. The statistical analysis is only one small aspect of the study. We have now added the software package used for analysis, and added some further detail to help clarify this section.

81 children were described as ‘other ethnicity’. If the focus is on SA children, I wonder if these should be excluded.

Although the focus was on SA children, the intervention was delivered to a multi-ethnic population, as would be the case if this was delivered in a wider population setting. We therefore think that including all the children is the correct approach for the main analysis. However, we have undertaken sensitivity analysis including only the SA children. For the outcomes of obesity and BMI z-score, the direction and size of effect were essentially the same. We have added a sentence to this effect within the paper.

The statement in the results section "Around one in five (n=115) were overweight or obese. This proportion was slightly higher in the control (21.7%) compared to the intervention (18.3%) schools, mainly due to a higher prevalence of overweight", is a little self explanatory.

The intention was that the difference between control and intervention groups was mainly due to the number of children who were overweight rather than the number who were obese. However, to make this simpler, we have now removed the last part of the sentence.

In table 2 there is missing data in some of the groups e.g. weight status for intervention is missing for 2 children - the total in each category should be made clear.

Figure 1 includes the number of valid measures for each variable, and we therefore did not include numbers of missing data in table 2. However, we have now updated table 2 to include the number of children we have data on for each characteristic/measure.

For the analysis focusing on 2 year follow-up. It states that "At follow up, the proportion of children who were overweight or obese had increased in all schools (from 7.3% to 9.9%, and from 12.8% to 19.1% for overweight and obese respectively)." It would be interesting to see here the proportion of children at follow-up, who were obese at baseline and follow-up, as there is some missing data. Also what proportion are underweight, normal, obese/overweight, rather than just obese as shown in table 3. And the differences between the intervention and control groups.

This is a public health intervention, where the aim was to prevent obesity at population level. Adding information on individual level change is therefore inappropriate. We have examined whether the difference in BMI z-score between control and intervention at follow up (adjusted for baseline values) differed in subgroups who were overweight/obese at baseline, compared to those who were non-overweight at baseline. The effect size was very similar in both groups (-0.14kg/m² in healthy weight versus -0.18kg/m² in overweight/obese group). We have not added this to the paper, but would be happy to do so if requested.

We have now changed table 3 into two separate tables: the first of which (labelled Table 3) shows the summary statistics for the anthropometric, physical activity and dietary measures in intervention and control groups at follow up. This table now includes the proportion of children in each of the four weight categories in the intervention and control groups at follow up. The other table (labelled Table 4) shows the adjusted odds ratios and mean differences between intervention and control groups for the outcomes considered. We have now added into this table the number of participants that were included in each of the models

For the continuous data in table 3, it wasn't clear what technique had been used here - was it ANCOVA to adjust for baseline values? It would be interesting to see the mean differences between the intervention and control group, rather than odd's ratios.

We have now updated the wording to make it clearer how we have adjusted for baseline values. We have added into the text the following wording:

“To adjust for baseline differences, we initially developed multiple linear regression models, which included the relevant baseline values of BMI, dietary factors or physical activity measures as covariates. Further models were then developed which also included potential confounders as covariates (age, sex, ethnicity).”

In table 3 it would also be useful to see the numbers in each analysis to show the extent of missing data.

Please see comment above: we have now added into Table 4 the number of participants include in each model.